# Effect of Hermaphrodite–Gynomonoecious Sexual System and Pollination Mode on Fitness of Early Life History Stages of Offspring in a Cold Desert Perennial Ephemeral

Jannathan Mamut [1,2], Junhui Cheng [3], Dunyan Tan [1,2,*], Carol C. Baskin [1,4,5] and Jerry M. Baskin [1,4]

1 College of Life Sciences, Xinjiang Agricultural University, Urumqi 830052, China; jinaiti@163.com (J.M.); carol.baskin@uky.edu (C.C.B.); jerry.baskin@yahoo.com (J.M.B.)
2 Ministry of Education Key Laboratory for Western Arid Region Grassland Resources and Ecology, College of Grassland Sciences, Xinjiang Agricultural University, Urumqi 830052, China
3 College of Resources and Environment, Xinjiang Agricultural University, Urumqi 830052, China; cjhgraymice@126.com
4 Department of Biology, University of Kentucky, Lexington, KY 40506, USA
5 Department of Plant and Soil Sciences, University of Kentucky, Lexington, KY 40546, USA
* Correspondence: tandunyan@163.com

**Abstract:** Gynomonoecy, the occurrence of both pistillate (female) and perfect (hermaphroditic) flowers on the same plant, has received little attention compared to gynodioecy and other plant sexual systems. *Eremurus anisopterus* is a perennial ephemeral in the cold desert of northwest China with a hermaphrodite–gynomonoecious sexual system in the same population. The primary aim of this study was to compare the early life history traits and inbreeding depression between progeny from pistillate and hermaphrodite flowers in hermaphrodites and gynomonoecious individuals. All of the traits of progeny from outcrossed pistillate flowers on gynomonoecious plants were significantly greater than for other pollination types. Selfing (vs. outcrossing) resulted in a decrease in all traits, indicating inbreeding depression (ID) during early life history stages of gynomonoecious and hermaphroditic plants. ID for seed mass, seed germination and seedling survivorship under water stress for pistillate flowers on gynomonoecious plants was significantly higher than it was for hermaphrodite flowers on both gynomonoecious and hermaphrodite plants. The advantage of the offspring of pistillate (vs. hermaphrodite) flowers may contribute to the maintenance of gynomonoecy in *E. anisopterus* in its cold desert sand dune habitat.

**Keywords:** early seedling growth; *Eremurus anisopterus*; hermaphrodite–gynomonoecious sexual system; inbreeding depression; offspring fitness; seed germination; seed mass

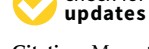



## 1. Introduction

Flowering plants exhibit a diversity of sexual systems that have combined male and female functions within and among individuals in seemingly every possible combination [1–3]. These sexual systems include hermaphroditism, monoecy, andromonoecy, gynomonoecy, dioecy, androdioecy and gynodioecy [3–5]. In recent decades, evolutionary biologists have been interested in the evolution and maintenance of diverse sexual systems because they might be an important mechanism in the promotion of outcrossing [6]. In flowering plants, this process is demonstrated by the transition from bisexual (hermaphrodite) to unisexual flowers and from bisexual to unisexual individuals [7]. Plants exhibit remarkable diversity in their sexual systems, which acts as a major driver of the genetic and evolutionary dynamics of flowering plants [8–10]. The evolution of plant sexual systems has been an important research issue since Darwin's studies [1]. Theoretical models suggest that the fitness consequences of selfing and outcrossing, the optimal allocation of resources to female and male functions, the genetic control of sex expression and environmental

conditions may be the result of the selective pressure of sex expression and can affect the sexual systems of flowering plants [9,11–13].

Gynomonoecy refers to the co-occurrence of female (pistillate) and perfect (hermaphrodite) flowers on the same plant [2,14,15]. Within populations, occurrence of the two types of flowers on gynomonoecious plants is due to genetic effects, including incomplete restoration of a male-sterile cytoplasm by one or many nuclear loci, and maternal effects, pollination system, resource allocation and herbivory effects [16]. Several hypotheses have been proposed to explain the evolution and maintenance of gynomonoecy. (1) As in other systems with unisexual flowers, the presence of the two flower types may permit the flexible allocation of resources to female and male reproductive functions in response to variations in environmental factors [17–19]. (2) Pistillate flowers may outcross more than perfect flowers. The outcrossing rate of pistillate ray florets of the radiate morph was significantly greater than that of perfect disc florets of either the radiate or nonradiate morph [20]. (3) The advantage of the pistillate flowers in aster heads lies in their attractiveness to pollinators [14]. (4) Pistillate flowers would be favored if perfect flowers were susceptible to biased predation by florivores [3,21,22]. In recent decades, most studies on gynomonoecy have focused on flowering phenology and female fitness [23], breeding system [24,25] floral sex ratios [14,26], sexual phenotypes [27], sex allocation and reproductive success [5], phylogeny [4,7,28] and the effects of biotic and abiotic factors of the environment on sex expression [3,29] on plants with gynomonoecy or the co-occurrence of gynomonoecious and other sexual systems. However, few studies have examined the effect of gynomonoecious sexual systems on progeny performance [30].

Theoretical and empirical work suggests that inbreeding depression (ID) will coevolve with the mating system, resulting in a positive correlation between outcrossing rates and inbreeding depression [31–33], and this is thought to play a key role in the evolution of plant breeding systems [11,34–37]. ID can vary between stages of the plant life cycle, and several studies have compared ID for seed production, seed mass, germination, survival and other plant life history traits in gynodioecious species (coexistence of females (male-sterile) and hermaphroditic individuals) [7,38,39]. However, other than studies on the eudicot genus *Silene* (*S. acaulis* [40] and *S. nutans* [41]), little is known about the effects of selfing versus outcrossing on the progeny fitness of gynomonoecious species.

*Eremurus anisopterus* Regel. (Xanthorrhoeaceae, formerly Liliaceae), is an early spring flowering perennial ephemeral that occurs in the cold deserts of Kazakstan and northwest China [42]. In China, *E. anisopterus* grows on fixed and semi-fixed sand dunes of the Gurbantunggut Desert of northern Xinjiang. Generally, the shoots of *E. anisopterus* begin to emerge aboveground in late March [15], and the temperature is low and fluctuates greatly during this period [43]. This species has a hermaphrodite–gynomonoecious sexual system [5]. About 30% of the pollen received by perfect (hermaphrodite) flowers of *E. anisopterus* is outcrossed pollen, while more than 60% of that received by pistillate (female) flowers is outcrossed pollen (J. Mamut, unpublished data). Furthermore, pistillate flowers produce larger seeds than hermaphroditic flowers in gynomonoecious plants, thereby compensating (in part) for the loss of pollen (male gamete) production [5]. Thus, we hypothesized that the progeny of pistillate flowers would have a greater performance in the early life history stages than those from hermaphrodite flowers in both hermaphrodites and gynomonoecious individuals. To test this hypothesis, we compared the progeny performance and inbreeding depression from different pollination types of pistillate and hermaphrodite flowers in both hermaphrodites and gynomonoecious individuals under stressful conditions.

## 2. Materials and Methods

### 2.1. Study Species and Seed Collection Site

Each flowering individual of *E. anisopterus* produces one large raceme with dozens of flowers. The racemes of gynomonoecious individuals have basal pistillate and distal perfect flowers. Flowering begins from late April to early May. Flowers do not have nectar

and are mainly pollinated by halictid bees and honeybees, which usually fly upwards when visiting receme inflorescences. Thus, pistillate flowers are more likely to produce outcrossed seeds [5]. *E. anisopterus* populations have both hermaphrodite (H) and gynomonoecious (Gm) plants, and the proportion of Gm individuals varies significantly from 2 to 17% among the populations [19].

Freshly matured seeds were collected from 900 plants in a large population of *E. anisopterus* growing on a cold desert sand dune in the Mosuowan region of the Gurbantunggut Desert (44°55′19.4″ N, 85°33′20.2″ E; 313 m a.s.l) in Xinjiang Uyghur Autonomous Region, northwest China. This region has a wide range of mean monthly temperatures. The coldest month is January ($-23$ °C), and the hottest month is July (33 °C), with a mean annual temperature of 7 °C. The annual precipitation (including rain and snow) is $147.3 \pm 8.6$ mm, with ~64% occurring in spring and summer. The monthly precipitation ranges from 5 (February) to 22 mm (July) (data from Mosuowan weather station, 1991–2010). Mean annual potential evaporation is 1942 mm [44].

*2.2. Pollination Type*

To examine the characteristics of seeds from different flower morph types and different types of pollination, we randomly labeled 900 hermaphroditic and gynomonoecious individuals in the population on 24–28 April 2014, and performed the following pollination treatments: (1) For hermaphrodite flowers, we tagged 300 hermaphroditic and 300 gynomonoecious individuals and established three pollination types (S, Op, Ou) of 100 hermaphrodite (H) and three pollination types of gynomonoecious (Gm) plants (see left and middle group of pollination types in Figure 1). Ten hermaphroditic (H) flowers were marked at the bud stage on each plant and enclosed in a mesh bag to avoid accidental pollination events. As soon as flowers opened, their stamens were cut, and the following hand pollinations were performed once stigmas became receptive: (a) open (designated as treatment Op), natural pollination without any floral manipulation; (b) self-pollinations were performed with pollen collected from other flowers of the same plant (S); or (c) outcross pollinations were made using a mixture of pollen from three hermaphrodites from the same population (Ou) (Figure 1). (2) For pistillate flowers, of the 300 selected Gm plants, we also marked 10 flowers in a basal position, using between four and seven pistillate (P) flowers on each plant, because *E. anisopterus* individuals generally produce between one and seven pistillate flowers. The pistillate flowers in each study group received three similar pollination types (see the group of pollination types on the right in Figure 1) as in the hermaphroditic plants. Flowers receiving treatments 2 and 3 were bagged after pollination to exclude any subsequent open pollination. Nonpollinated flowers were removed to keep the number of flowers per mother plant similar and to avoid resource allocation to nontarget flowers.

To easily distinguish the results of different plant, flower and pollination types, we used three-letter acronyms to refer to specific groups of results. The three letters of the acronym refer to the plant type (hermaphrodite (H) vs. gynomonoecious (Gm)), the flower type (hermaphroditic flowers of H, hermaphroditic and pistillate flowers of Gm) and the pollination type (open pollination, selfing and outcrossing) (Figure 1). For example, GmP-S refers to seeds produced by a pistillate flower on a gynomonoecious plant that was self-pollinated.

*2.3. Seed and Seedling Traits Measured*

For seeds produced using the above treatments, we measured seed set, seed mass, embryo growth, germination percentage and seedling survival and growth.

2.3.1. Seed Set

Seed set was measured by randomly sampling one fruit each from 20 individuals of each pollination type, and the seed set per fruit was determined for all treatments.

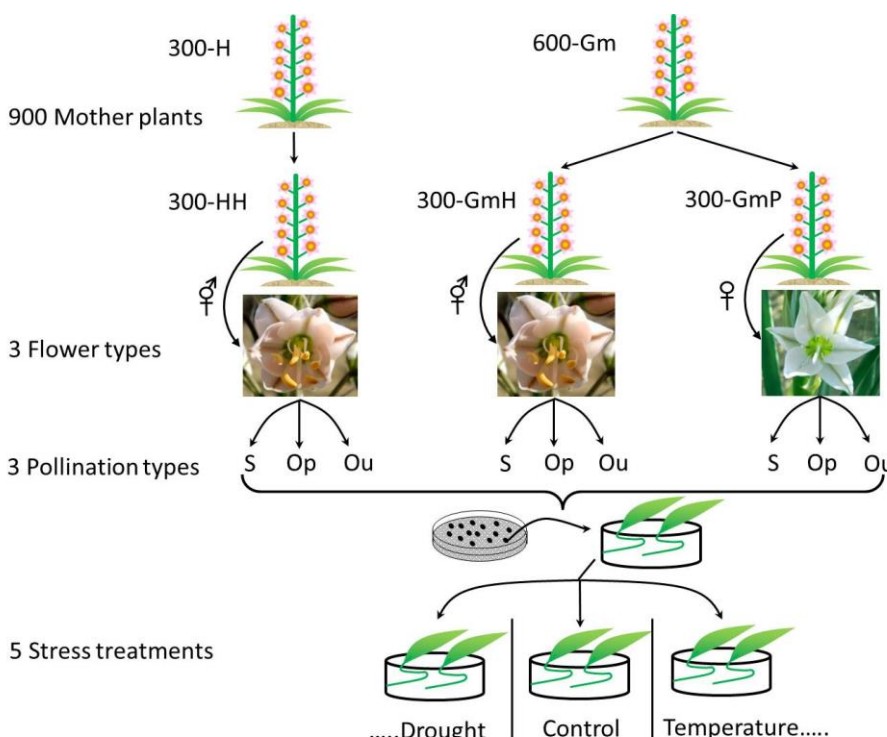

**Figure 1.** Schematic diagram showing the sequence of pollination types and stress treatments. H—hermaphrodite plants; Gm—gynomonoecious plants; HH—hermaphrodite flowers on hermaphroditic plants; GmH—hermaphrodite flowers on gynomonoecious plants; GmP—pistillate flowers on gynomonoecious plants. S—selfed; Op—open pollinated; Ou—crossed.

### 2.3.2. Seed Mass

Seed mass was measured by randomly sampling 30 seeds (one seed per fruit from each of 30 plants) from each pollination type, and the individual seeds were weighed using a Sartorius BS210S electronic balance (0.0001 g).

### 2.3.3. Phenology of Embryo Growth

Embryos in seeds of *E. anisopterus* are underdeveloped, and consequently they must grow inside the seed before radicle emergence (germination) [45]. To determine the initial length (size) of the embryo, 25 seeds from each pollination type were placed on moist filter paper in Petri dishes at room temperature on 27 June 2014. After 24 h, each seed was cut open with a razor blade, and embryo length and seed length (E:S ratio) were determined using a dissecting microscope equipped with a micrometer.

To monitor the phenology of embryo growth, 30 freshly mature seeds of each pollination type were placed in 10 fine-mesh nylon bags and buried in clay pots (30 cm diameter, 30 cm height) to a depth of 3 cm in sand from the natural habitat of *E. anisopterus*. The pots were buried in soil (top of pot level with soil surface) in an experimental garden on the campus of Xinjiang Agricultural University, Urumqi, on 27 June 2014. Sand in the pots was kept moist throughout the monitoring period. After burial, 25 or 30 seeds (some seeds decayed during burial) were removed from one haphazardly selected bag every 15 d until 26 October 2014 and the E:S ratio was determined as described above. Sand temperatures at a depth of 3 cm were recorded at 2 h intervals throughout the burial period using a Tiny Tag data logger (Model Micro Lite LITE5016, Fourier Technologies, Beijing, China). Daily mean, maximum and minimum temperatures were calculated from these data.

### 2.3.4. Seed Germination

One hundred freshly mature seeds of each of the nine levels of pollination types were sown on wet filter paper with 12 cm diameter Petri dishes and cold stratified at

5/2 °C (12/12 h) for 8 weeks in darkness [45]. After cold stratification, four replicates of 25 seeds each were transferred to 9 cm diameter Petri dishes under green light and incubated in darkness at 15/2 °C for 30 days. Seeds incubated in darkness were checked after 30 d for germination, and distilled water was added every 7 d (in green light). After the germination trials were complete, non-germinated seeds were tested for viability, and germination percentages were calculated based on number of viable seeds [45].

### 2.3.5. Seedling Growth and Survival

To compare the early growth of seedlings from the nine pollination types, we used seeds that had been stored dry at room conditions for 6 months. One hundred seeds from each of the nine pollination types were placed on moist sand in 12 cm diameter Petri dishes and cold stratified at 5/2 °C for 8 weeks in darkness. After treatment, four replicates of 25 seeds were transferred under green light to new 12 cm diameter Petri dishes and incubated in darkness at the 15/2 °C temperature regime for 4 weeks. These procedures have been shown to break seed dormancy, thus allowing for the measurement of subsequent seedling traits.

In addition to growing seedlings in the 15/2 °C optimal condition, we also examined seedlings' response to two types of stressful conditions often found in the natural environment of this species—specifically, high temperature and low soil water content.

### Temperature

For these treatments, we transferred four replicates of 10 3-week-old seedlings from moist filter paper in Petri dishes into a hydroponic system. In this system, individual seedlings were placed into 1 cm diameter holes (1 seedling per hole) bored into round pieces of styrofoam (1 cm thick templates) that covered the Petri dishes (12 cm diameter, 2.5 cm deep) containing 80 mL of 1/2 strength Hoagland nutrient solution. Roots of the seedlings reached into the nutrient solution, which was replaced with fresh Hoagland solution 2 weeks after the initial transplantation, at the mid-point of the experiment. Using this setup, we incubated two groups of seedlings at daily (12/12 h) temperature regimes of 5/2 and 20/10 °C in light for 28 d, representing suboptimal, and superoptimal temperature, respectively. The optimum temperature regime for the seed germination of *E. anisopterus* is 15/2 °C [45]. Thus, we grew an additional group of seedlings at 15/2 °C to serve as the control. Seedlings were arranged haphazardly, and all Petri dishes were rotated several times weekly to reduce the possibility of positional effects. Seedling survival was monitored at 1-week intervals and height, root length and mass of seedlings were measured only at the end of the experiment.

### Water Stress

The design used to grow seedlings under water stress was very similar to the one used to test the effect of temperature stress on seedling growth. Three replication of 10 3-week-old seedlings were transferred from the moist filter paper into Petri dishes containing 0 (control, 1/2 strength Hoagland nutrient solution only (HNS)), −0.15 and −0.51 MPa PEG (HNS+ PEG) solutions at 15/2 °C for 28 d [46]. Seedling survival was monitored at 1-week intervals and height, root length and mass of seedlings were measured only after 28 d.

### 2.4. Effect of Inbreeding Depression on Seed Set, Mass, Germination and Seedling Growth and Survival

Using the measurements from different pollination types, we calculated the inbreeding depression (δ) for individual traits (seed set, seed mass, embryo growth, seed germination and seedling growth and survival) using the equation $\delta = 1 - (W_s/W_o)$, where $W_s$ and $W_o$ were the measurement of individual fitness traits for selfed and out-crossed progeny, respectively. Thus, $(W_s/W_o)$ is relative fitness [47,48]. In this study, $W_s$ was lower than $W_o$ in all cases; thus, our measurement of inbreeding depression was always positive.

*2.5. Multiplicative Fitness of Early Life History Traits*

The effects of inbreeding are cumulative (multiplicative) across the life cycle [48]. Thus, the cumulative fitness of the early life history stages of *E. anisopterus* was estimated by calculating the product of relative fitness ($W_s/W_o$) for seed set, seed mass, embryo growth, germination percentage and seedling survival/growth (cumulative relative fitness, CRF); 1—CRF = total ID for these three stages of *E. anisopterus*.

All calculations for inbreeding depression were conducted separately for the three types of flowers: hermaphrodite and pistillate flowers on gynomonoecious plants, and hermaphrodite flowers on hermaphrodite plants.

*2.6. Data Analysis*

A linear-mixed model was used to explore the effects of explanatory variable (plant type, flower type and pollination type) on response variable (seed set, seed mass, embryo growth and seed germination). In the linear-mixed model, fixed effects were detected by the main and all potential interactive effects among these explanatory variables except for the interactive effect between plant and flower types, since flower type was not fully crossed with plant type. Plant type was also considered a random factor nested with individuals, which was randomly sampled. The linear-mixed model was conducted by "lme" function in "nlme" packages in R software [49]. If fixed factors had significant main or interactive effects in the linear-mixed model, we further used the "lsmean" function in "lsmean" package to conduct multiple comparisons [50].

To express the relative importance of each explanatory variable on response variables, we firstly calculated adjusted $R^2$ for the whole linear-mixed model. Then, the main and interactive effects of each explanatory variable were removed from the whole model, and their adjusted $R^2$ was calculated as suggested by de Vries et al. [51]. The relative importance of each explanatory variable was depicted by its adjusted $R^2$ calculated from the removed model [51]. In particular, the adjusted $R^2$ was calculated by the "r. squared LR" function in "MuMIn" packages [52]. Finally, we used the linear regression to examine the relationship between seed mass and seed germination and seedling traits.

## 3. Results

*3.1. Seed Traits*

The linear-mixed model results revealed that flower type and pollination type significantly affected all measured traits except embryo growth (however, flower type significantly affected the final E:S ratio) (Table 1). There were no significant effects of plant type on seed mass or seed germination, nor was there a significant interaction between flower type and pollination type or between plant type and pollination type. However, plant type significantly affected seed set (Table 1). More specifically, flowers that were cross-pollinated produced more (Table 1; Figure 2A) and larger (Table 1; Figure 2B) seeds than self-pollinated flowers, and the open pollinated flowers exhibited intermediate trait values in all cases. The germination of cross-pollinated flowers was significantly higher (Table 1; Figure 2C) than that of self-pollinated flowers except HH.

*3.2. Phenology of Embryo Growth*

The initial E: S ratio (28 June) for fresh seeds of each pollination type ranged from $0.707 \pm 0.009$ for selfed seeds of HH to $0.735 \pm 0.018$ for outcrossed seeds of GmP, and differences among pollination types were not significant (Table 1, Figure 3).

Embryos grew only a little during summer 2014; however, between 11 September and 26 October 2014, during which time mean daily maximum and minimum sand temperatures were 14.8 and 4.6 °C, respectively, embryos grew rapidly (Figure 3A). The final E: S ratio (26 October) ranged from $0.868 \pm 0.007$ for HH-S to $0.899 \pm 0.007$ for GmP-Ou, and the differences among the flower types were significant (Table 1). Embryos from GmP-Ou and GmP-Op reached the critical E:S ratio required for germination on 26 October 2014, at which time 2% of the seeds in the bags had germinated.

**Table 1.** Results of the linear mixed model showing the effect of plant type (hermaphroditic vs. gynomonoecious), flower type (hermaphroditic flowers of H, hermaphroditic and pistillate flowers of Gm), pollination type (open pollination, selfing and outcrossing) and their interactions in terms of seed set, seed mass and seed germination. Effects with $p < 0.05$ are shown in boldface type.

| Source of Variation | d.f. | F | p |
|---|---|---|---|
| **Seed set** | | | |
| Plant type (PT) | 1, 19 | 20.799 | **0.0002** |
| Flower type (FT) | 1, 133 | 86.986 | **<0.0001** |
| Pollination type (PT′) | 2, 133 | 11.448 | **<0.0001** |
| FT * PT′ | 2, 133 | 10.258 | **0.0001** |
| PT * PT′ | 2, 133 | 0.0241 | 0.976 |
| **Seed mass** | | | |
| Plant type (PT) | 1, 29 | 2.215 | 0.147 |
| Flower type (FT) | 1, 203 | 44.407 | **<0.0001** |
| Pollination type (PT′) | 2, 203 | 32.104 | **<0.0001** |
| FT * PT′ | 2, 203 | 0.967 | 0.382 |
| PT* PT′ | 2, 203 | 1.359 | 0.259 |
| Initial E:S ratio | | | |
| Plant type (PT) | 1, 218 | 0.620 | 0.433 |
| Flower type (FT) | 1, 218 | 0.300 | 0.255 |
| Pollination type (PT′) | 2, 218 | 0.130 | 0.324 |
| FT * PT′ | 2, 218 | 0.010 | 0.987 |
| Final E:S ratio | | | |
| Plant type (PT) | 1, 218 | 1.230 | 0.268 |
| Flower type (FT) | 1, 218 | 5.110 | 0.024 |
| Pollination type (PT′) | 2, 218 | 1.580 | 0.209 |
| FT * PT′ | 2, 218 | 0.920 | 0.401 |
| **Seed germination** | | | |
| Plant type (PT) | 1, 3 | 9.320 | 0.055 |
| Flower type (FT) | 1, 21 | 25.329 | **0.0001** |
| Pollination type (PT′) | 2, 21 | 24.533 | **<0.0001** |
| FT * PT′ | 2, 21 | 1.4868 | 0.249 |
| PT* PT′ | 2, 21 | 0.8807 | 0.429 |

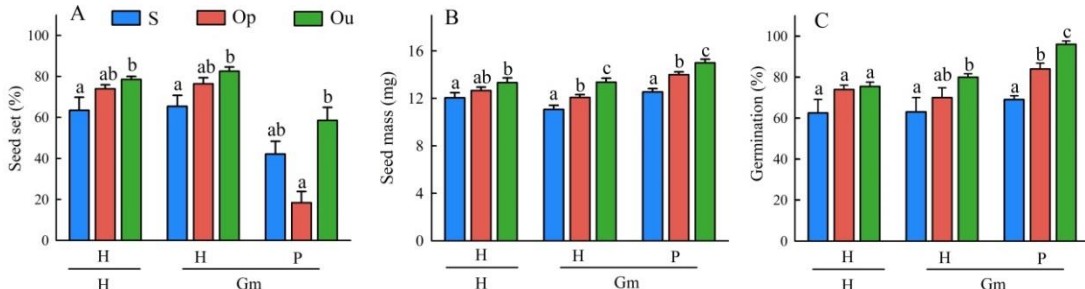

**Figure 2.** Mean (± SE) seed set (**A**), seed mass (**B**), and germination percentage (**C**) of seeds from selfed (S), open pollinated (Op) and crossed (Ou) flowers of *Eremurus anisopterus*. HH, hermaphrodite flowers on hermaphrodite plants; GmH, hermaphrodite flowers on gynomonoecious plants; GmP, pistillate flowers on gynomonoecious plants. Different letters above each group indicate significant differences between pollination types within each flower type ($p < 0.05$).

*3.3. Seedling Traits*

Flower type, pollination type and stress treatment had significant effects on seedling survival and biomass (Table 2). There were no significant effects of plant type on survival, but plant type significantly affected biomass. However, flower type, pollination type and stress treatment interactions ($p < 0.05$) had no significant effects on seedling survival or flower type, and pollination type and stress treatment interactions ($p < 0.05$) had no significant effects on biomass (Table 2). Seedling survival and biomass were higher for the

control than for seedlings exposed to stress treatments. Selfed progeny had significantly lower survivorship and lower biomass than outcrossed progeny incubated in –0.51 MPa in three flower types except the GmH in biomass. Seedlings from the two groups of hermaphroditic flowers (from the H and Gm plants) were similar in terms of survival and biomass (Figure 4).

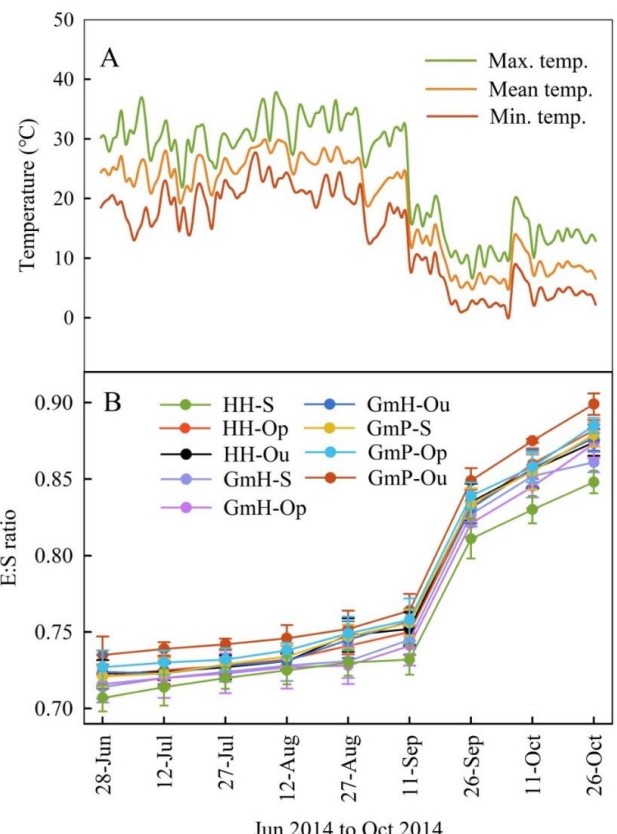

**Figure 3.** Daily maximum, daily minimum and daily mean sand temperatures at a depth of 3 cm (**A**) and phenology of embryo growth in seeds from each pollination type in *Eremurus anisopterus* and (**B**) in the experimental garden. Error bars in (**B**) are ±s.e.

**Table 2.** Results of linear mixed model showing the effect of plant type (hermaphroditic vs. gynomonoecious), flower type (hermaphroditic flowers of H, hermaphroditic and pistillate flowers of Gm), pollination type (open pollination, selfing and outcrossing), stress treatment (control, 2 temperatures, and 2 water stresses) and their interactions in terms of seedling survival and seedling biomass. Effects with $p < 0.05$ are shown in boldface type.

| Source of Variation | Survival | | | Biomass | | |
|---|---|---|---|---|---|---|
| | d.f. | F | p | d.f. | F | p |
| Plant type (PT) | 1, 2 | 4.939 | 0.1563 | 1, 19 | 44.918 | **<0.0001** |
| Flower type (FT) | 1, 86 | 10.623 | **0.0016** | 1, 817 | 131.391 | **<0.0001** |
| Pollination type (PT′) | 2, 86 | 26.427 | **<0.0001** | 2, 817 | 126.087 | **<0.0001** |
| Stress treatment (ST) | 4, 86 | 124.124 | **<0.0001** | 4, 817 | 333.552 | **<0.0001** |
| FT * PT′ | 2, 86 | 1.213 | 0.3023 | 2, 817 | 5.477 | **0.0043** |
| FT * ST | 4, 86 | 3.896 | **0.0059** | 4, 817 | 2.831 | **0.0238** |
| CT * ST | 8, 86 | 3.53 | **0.0014** | 8, 817 | 1.402 | 0.1918 |
| PT * PT′ | 2, 86 | 3.049 | 0.0525 | 2, 817 | 6.697 | 0.0013 |
| PT * ST | 4, 86 | 3.721 | **0.0077** | 4, 817 | 0.074 | 0.9901 |
| FT * PT′ * ST | 8, 86 | 0.257 | 0.9778 | 8, 817 | 1.199 | 0.2967 |
| PT * PT′ * ST | 8, 86 | 0.426 | 0.9023 | 8, 817 | 0.058 | 0.9999 |

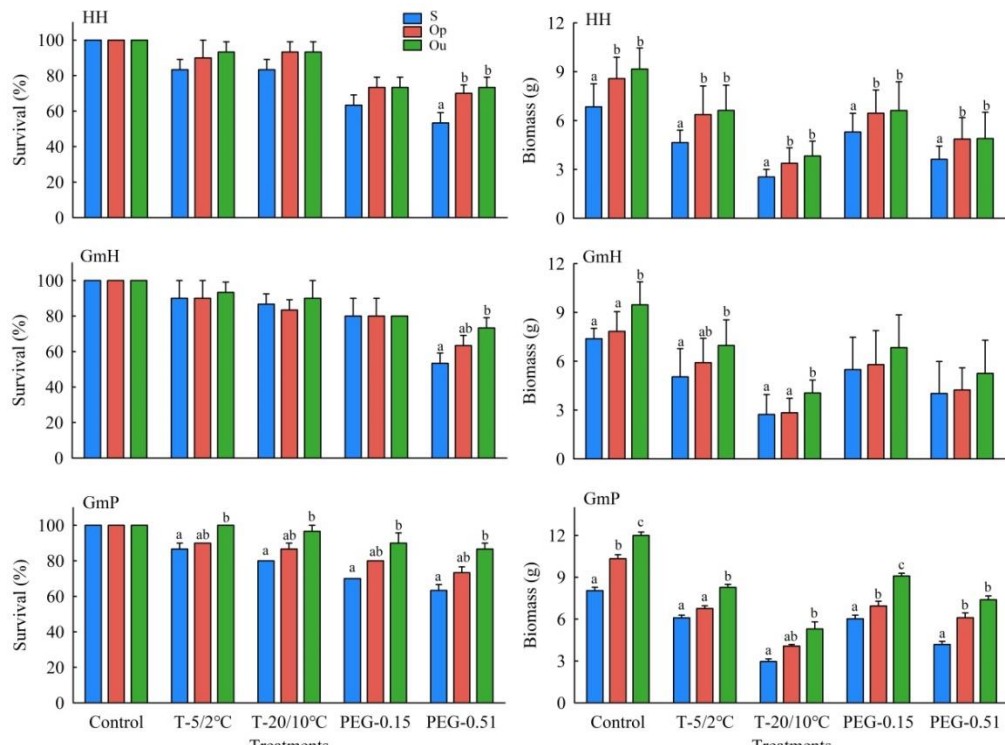

**Figure 4.** Mean (±SE) seedling survival (left) and biomass (right) of offspring from seeds of selfed (S), open pollinated (Op) and crossed (Ou) flowers of *Eremurus anisopterus* under different stress conditions. HH, hermaphrodite flowers on hermaphrodite plants; GmH, hermaphrodite flowers on gynomonoecious plants; GmP, pistillate flowers on gynomonoecious plants. Different letters above each group indicate significant differences between pollination types within each treatment ($p < 0.05$). Marginally significant differences are not shown here.

### 3.4. Effect of Seed Mass on Germination and Seedling Traits

Germination percentage of different pollination type seeds from H and P flowers from hermaphrodite and gynomonoecious plants was positively correlated with seed mass ($R^2 = 0.93$, $p < 0.001$) (Figure 5). Outcrossed seeds from pistillate flowers on gynomonoecious plants were significantly larger than those of selfed seeds from gynomonoecious and hermaphroditic plants. ANCOVA showed that germination percentage was independent of seed mass (Table 3).

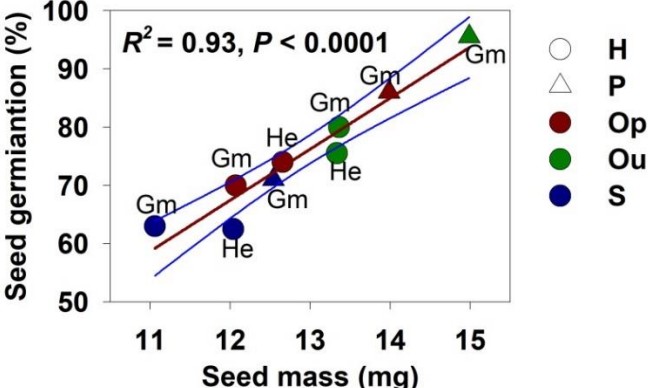

**Figure 5.** Relationship between seed mass and seed germination in *Eremurus anisopterus*. Red lines show linear regressions, and blue lines show ± 95% confidence intervals. He—hermaphrodite plants; Gm—gynomonoecious plants; H—hermaphrodite flowers P—pistillate flowers; S—selfed—Op—open pollinated; Ou—crossed.

**Table 3.** Analysis of covariance (ANCOVA) of the effect of seed size on germination.

| Source of Variation | d.f. | MS | F | p |
|---|---|---|---|---|
| Pollination type | 8 | 202.272 | 3.239 | 0.011 |
| Seed mass | 1 | 49.132 | 0.787 | 0.383 |
| Error | 26 | 62.456 | | |
| $R^2 = 0.588$ | | | | |

Seed mass tended to affect seedling survival and biomass, and its effect varied under stressful conditions. However, there was no significant effect in the control and PEG —0.15. The survival and biomass of selfed progeny from larger outcrossed seeds from pistillate flowers on gynomonoecious plants (which were larger than selfed seeds from gynomonoecious and hermaphroditic plants) was significantly higher than that of selfed seeds from both gynomonoecious and hermaphroditic plants (which were smaller than outcrossed seeds from female flowers on gynomonoecious plants), especially for outcrossed progeny from pistillate flowers on gynomonoecious plants under PEG —0.51 MPa (Figure 6).

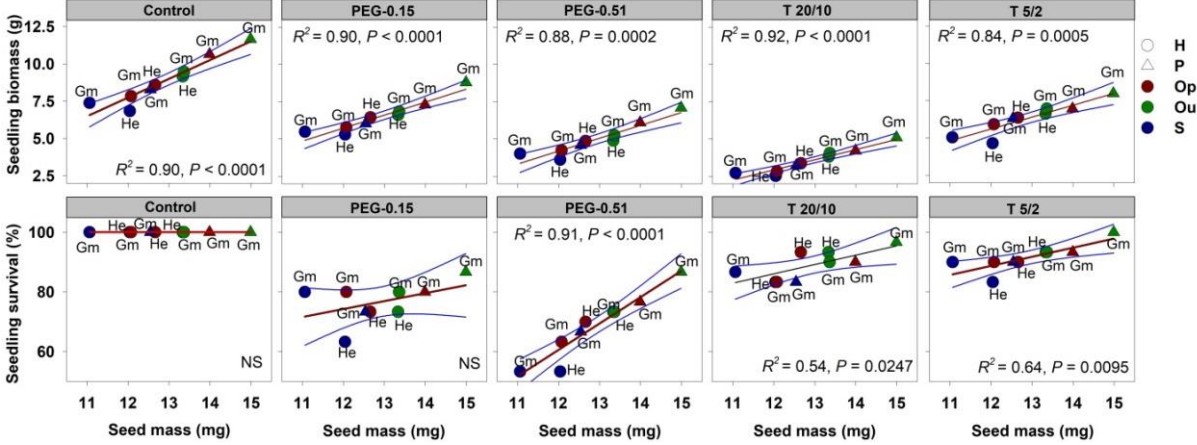

**Figure 6.** Relationship between seed mass and seedling survival and biomass in *Eremurus anisopterus*. Red lines show linear regressions, and blue lines show ±95% confidence intervals. NS, no significant difference. Abbreviations as in Figure 5.

### 3.5. Effect of Inbreeding Depression on Seed and Seedling Traits

Inbreeding depression for seed mass and germination was much stronger for GmP than for GmH and HH. However, seed set and embryo growth did not differ in inbreeding depression among three flower types. All of the germinated seeds survived in the control, suggesting that inbreeding did not affect survival under benign conditions. Under suboptimal and superoptimal temperatures (5/2 °C and 20/10 °C), seedling survivor exhibited weak inbreeding depression. However, under water stress (−0.51 Mpa) seedling survivorship exhibited a much stronger inbreeding depression (Table 4).

Total ID for the early growth stages for GmH, GmP and HH ranged from 0.53 to 0.65, 0.59 to 0.80 and 0.46 to 0.64, respectively (Table 5).

**Table 4.** Inbreeding depression for seed set, seed mass, embryo growth, germination and seedling survival of *Eremurus anisopterus* under different stress conditions. GmH—hermaphrodite flowers on gynomonoecious plants; GmP—pistillate flowers on gynomonoecious plants; HH—hermaphrodite flowers on hermaphrodite plants. S-C; seedling survivorship in control (1/2 strength Hoagland nutrient solution); S-T—seedling survivorship under temperature stress; S-W—seedling survivorship under water stress. Different letters within a row indicate significant differences (Tukey's HSD, $p < 0.05$).

| Treatment | Flower Type | | | *F* | *p* |
|---|---|---|---|---|---|
| | **GmH** | **GmP** | **HH** | | |
| Seed set | 0.188 ± 0.082 [a] | 0.188 ± 0.073 [a] | 0.164 ± 0.211 [a] | 0.010 | 0.990 |
| Seed mass | 0.102 ± 0.006 [a] | 0.165 ± 0.007 [b] | 0.097 ± 0.009 [a] | 26.219 | 0.000 |
| Initial E:S ratio | 0.024 ± 0.006 [a] | 0.026 ± 0.013 [a] | 0.021 ± 0.008 [a] | 0.061 | 0.941 |
| Final E:S ratio | 0.019 ± 0.002 [a] | 0.024 ± 0.006 [a] | 0.017 ± 0.004 [a] | 0.502 | 0.607 |
| Germination | 0.205 ± 0.031 [a] | 0.256 ± 0.017 [b] | 0.202 ± 0.046 [a] | 16.523 | 0.003 |
| S-C | 0.000 ± 0.000 [a] | 0.000 ± 0.000 [a] | 0.000 ± 0.000 [a] | — | — |
| S-T 5/2 °C | 0.086 ± 0.016 [a] | 0.088 ± 0.003 [a] | 0.062 ± 0.003 [a] | 2.353 | 0.176 |
| S-T 20/10 °C | 0.074 ± 0.016 [a] | 0.077 ± 0.021 [a] | 0.052 ± 0.003 [a] | 0.773 | 0.503 |
| S-W —0.15 Mpa | 0.129 ± 0.012 [a] | 0.157 ± 0.007 [b] | 0.125 ± 0.010 [a] | 13.251 | 0.001 |
| S-W —0.51 Mpa | 0.268 ± 0.016 [a] | 0.294 ± 0.005 [b] | 0.271 ± 0.030 [a] | 12.933 | 0.003 |

**Table 5.** Total inbreeding depression (TID) for seed set, seed mass, final E:S ratio, seed germination and seedling survival/growth of *Eremurus anisopterus* under different environmental stresses. GmH—hermaphrodite flowers on gynomonoecious plants; GmP—pistillate flowers on gynomonoecious plants; HH—hermaphrodite flowers on hermaphrodite plants. S-C; seedling survivorship in control (1/2 strength Hoagland nutrient solution); S-T—seedling survivorship under temperature stress; S-W—seedling survivorship under water stress.

| Treatment | GmH | GmP | HH |
|---|---|---|---|
| S-C | 0.529 | 0.719 | 0.567 |
| S-T 5/2 °C | 0.604 | 0.714 | 0.597 |
| S-T 20/10 °C | 0.605 | 0.593 | 0.627 |
| S-W —0.15 Mpa | 0.566 | 0.772 | 0.464 |
| S-W —0.51 Mpa | 0.648 | 0.798 | 0.639 |

## 4. Discussion

All traits of outcross seeds from pistillate flowers on gynomonoecious plants were significantly higher than they were for those of the other pollination types. Thus, our hypothesis that the life history traits of pistillate flowers are more fit and that ID would be higher for the progeny of pistillate flowers than for those of hermaphrodite flowers in both hermaphrodites and gynomonoecious individuals is supported.

Seeds produced by different pollination types may affect the fitness of the post-germination traits of the progeny [47,53,54]. In natural populations of *E. anisopterus*, pistillate flowers received more outcross pollen than perfect flowers, resulting in 60% of the seeds being outcrossed, while perfect flowers received more pollinator visits than pistillate flowers but had a higher proportion of geitonogamy, resulting in 30% of the seeds being outcrossed (J. Mamut, unpublished data). Hence, the presence of pistillate flowers on gynomonoecious plants could help reduce selfing rates, in support of the outcrossing hypothesis of gynomonoecy.

Seed mass is important in the life history of plants [55–57], because it has a strong influence on seed germination, seedling emergence, survivorship, seedling size and seedling competitive ability [56,58]. Various studies have shown that seeds from female flowers of gynodioecious species are larger than those of perfect flowers [30,59]. Perfect flowers of *E. anisopterus* produced significantly more seeds than pistillate flowers [5]. However, the mass of individual outcrossed seeds from pistillate flowers on gynomonoecious plants

was significantly greater than that from the other pollination types (Figure 2B, Table 1), suggesting that pollination type had a significant effect on seed mass and that the pistillate flowers on plants of this gynomonoecious species increase female function via seed quality.

Seeds of *E. anisopterus* are dispersed in late June and, at this time, the embryos of all pollination types of seeds are underdeveloped. The embryos began to elongate in summer, but they grew slowly. Between mid-September and late October, embryos grew rapidly and reached their critical length for radical emergence (germination) [5]. The critical E: S ratio of GmP-Ou was significantly greater than that of other pollination types, indicating that embryos of outcrossed seeds of pistillate flowers (GmP-Ou) elongate more than those of selfed (GmH-S, GmP-S and HH-S) embryos. Ours is the first study on the effect of flower cross-type on embryo growth.

Some species described as gynodioecious are truly gynomonoecious–gynodioecious, with three phenotypes: females, perfect-flowered hermaphrodites and gynomonoecious individuals [41,60]. In the gynomonoecious–gynodioecious species *Silene acaulis,* seeds from three flower types (pistillate flowers on females, pistillate flowers on gynomonoecious and perfect flowers on gynomonoecious plants) did not differ significantly in germination percentage in either the greenhouse or field [40]. In another gynomonoecious–gynodioecious species, *S. nutans*, Dufay et al. [41] observed that crossed seeds of hermaphrodites germinated to a higher percentage than selfed seeds of both gynomonoecious and hermaphrodite plants. In *E. anisopterus*, the germination percentage and seedling performance of larger outcrossed seeds from pistillate flowers on Gm plants were significantly higher than those of selfed seeds from hermaphrodite flowers on both Gm and H plants [56,61,62].

The relative frequency of selfing and outcrossing influences the offspring fitness and genetic diversity of flowering plants [63]. The production of pistillate flowers enhances the opportunity for outcrossing. Many studies of gynodioecious species found that the offspring of females germinates to a higher percentage and grows larger than the offspring of hermaphrodites [38,64,65]. In *E. anisopterus*, pollination type had a significant effect on early seedling growth and survival. Under stressful conditions, seedlings from cross-pollinated pistillate flowers on gynomonoecious plants grew and survived better than those of seedlings from the other pollination types (Figure 4). This suggests that seedlings from female flowers on gynomonoecious plants have more potential to grow and survive in the unpredictable environment (with regard to timing and amount of precipitation) of the cold deserts in Central Asia in early spring than those from perfect flowers of gynomonoecious or hermaphroditic plants.

Inbreeding depression is thought to be a major selective factor in the evolution of reproductive system diversity in flowering plants, particularly in maintaining outcrossing in spite of the automatic gene transmission advantage of self-fertilization [66–68]. For *E. anisopterus*, we found inbreeding depression in gynomonoecious and hermaphroditic plants for different life cycle traits. A decrease in both seed mass and germination percentage resulting from selfing in *E. anisopterus* is consistent with the results reported for gynomonoecious–gynodioecious species *S. acaulis* [40], *Dianthus sylvestris* [69] and *S. nutans* [41]. Furthermore, we found significant inbreeding depression in GmH, GmP and HH for different life cycle stages. ID for seed mass, germination and seedling survivorship under water stress for GmP was significantly higher than ID for GmH and HH. However, there was no difference between GmH and HH, suggesting that flower type had a significant effect on inbreeding depression. ID may or may not increase with stress (See Table 1 in [48] for references on the effect of competition and physical environment on inbreeding depression in plants). For *E. anisopterus*, inbreeding depression was found for the survivorship of seedlings under environmental stress, and especially for those under water stress. The outcrossed progeny of *E. anisopterus* had higher survivorship than selfed progeny, suggesting that, for this species, recessive alleles are more deleterious under stressful conditions [39].

## 5. Conclusions

Several hypotheses have been proposed to explain the evolution and maintenance of gynomonoecy. They suggest that this sexual system may enhance outcrossing [5,21,69], avoid pollen–pistil interference [14,26], act as a defense against herbivores [3,21,23], enhance attractiveness to pollinators [21,70] and/or permit flexible resource allocation [17]. Our study shows that total ID was higher for the progeny of pistillate flowers than for those of hermaphrodite flowers in hermaphrodites and gynomonoecious individuals across the early history stages. Seed mass, seed germination and seedling survival/growth of outcrossed seeds from pistillate flowers on gynomonoecious plants were significantly higher than they were for those of the other pollination types. Thus, the female flower has an advantage over the hermaphrodite flower in *E. anisopterus*. These results are consistent with the outcrossing-benefit hypothesis for gynomonoecy, which proposes that female flowers of gynomonoecious plants can partially avoid inbreeding depression by favoring outcrossing rates [21,29].

**Author Contributions:** J.M.: conceptualization (equal), investigation (equal), methodology (equal), software (equal), writing—original draft (lead); J.C.: conceptualization (supporting), methodology (supporting), software (equal); D.T.: conceptualization (equal), methodology (equal), writing—review and editing (lead); C.C.B.: conceptualization (equal), methodology (equal), writing—review and editing (lead); J.M.B.: conceptualization (equal), methodology (equal), writing—review and editing (lead). All authors have read and agreed to the published version of the manuscript.

**Funding:** This work was supported in part by National Natural Science Foundation of China (31960053), Tianshan Youth Program of Xinjiang Uygur Autonomous Region (2019Q019) and National Natural Science Foundation of China (U1603231).

**Institutional Review Board Statement:** Not applicable.

**Informed Consent Statement:** Not applicable.

**Data Availability Statement:** Not applicable.

**Acknowledgments:** We thank Shu-Mei Chang from University of Georgia for valuable comments that helped us to improve the manuscript.

**Conflicts of Interest:** The authors declare no conflict of interest.

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
