# Peer review of "Effect of Hermaphrodite–Gynomonoecious Sexual System and Pollination Mode on Fitness of Early Life History Stages of Offspring in a Cold Desert Perennial Ephemeral"

_diversity, doi:10.3390/d14040268_

Round 1
Reviewer 1 Report
Review of MS « Effect of hermaphrodite-gynomonecious sexual system and pollination mode on fitness of early life history stages of offspring in a cold desert perennial ephemeral
Overall comments
The current study aims at investigating how pollen quantity and origin (outcross vs self pollen) affect seed performance, by also taking into account both the flower type (female vs hermaphroditic) and plant type (gynomonoecious vs. hermaphroditic). I think the idea behind this is to understand whether avoidance of inbreeding depression in female flowers may explain polymorphism in sexual types of flowers and individuals but the link between the general knowledge about sexual systems and the exact expectations is not clearly explained in the introduction. Besides, I strongly disagree with the expectations exposed at the end of the introduction and the way the authors decide to test for their hypothesis (see below)
Overall, I found the present manuscript quite unreadable. The introduction does not allow the reader to understand the several hypotheses that can be tested with the experiments (and I suspect them to be based on faulty reasoning). Both the Material and Methods and the Result sections contain multiple mistakes or inconsistencies making it painful to read.
The study itself seems to be well conducted and contains interesting data. But the authors need to rewrite their manuscript, in order to (i) rethink their working hypothesis, (ii) better explain how their results shed light about what we know about the evolution of plant sexual systems (instead of vaguely arguing that inbreeding depression is a major factor of everything) and (iii) make sure that no mistakes / inconsistencies remain in their text if they want the reader to understand what they have done.
I have tried to illustrate these issues below, by pointing some of the aspects that need to be improved. If I must be honest, after having struggled to follow the experiment and to understand some of the graphs, I did not invest much time in thinking about the discussion. Such activity is possible if we understand the hypothesis to be tested and the results.
Specific comments
Introduction
At several locations of the text, references are wrongly cited. Here are a few examples:
Line 39 : the cited references are (i) a phylogenetic study that explores the transition between several sexual systems and (ii) one descriptive study of gynomonecy. This does not fit with the sentence “explore ecological and genetic requirements for the evolution of sex expression in flowering plants”! MANY papers (based on models, and reviews) can be cited here. And it could be worth explaining in shord the main conclusions of these studies
Line 57: two of the references cited to illustrate that inbreeding depression is a major factor that has been identified to explain the evolution of mating system are case studies on one species each. There is a huge list of theoretical and review papers that discussed the effect of ID. And the question of mixed mating system (i.e. is the maintenance of intermediate selfing rates a stable strategy and in which conditions) is 1) not directly linked with the current study in my opinion and 2) has been treated in a rich literature that is not cited here.
Otherwise, I found the introduction sometimes vague, with several false statements. For example:
Lines 33-36 : I cannot understand this sentence. This is, I think, grammatically wrong. I don’t understand what “vital source of genetic variation” means, neither how the evolution of diverse sexual system could be “a mechanism for outbreeding”
Line 41: I would not say that gynomonecy is “an important step in the evolutionary pathway from hermaphrodites to monoecy in angiosperms. It has been found to a pathway in the Asteraceae family. But to my knowledge, that’s it. (and I am not sure to understand why the paper by Charleworth and Charlesworth is cited here..)
Lines 43/44: I think the sentence is grammatically incorrect. And completely uninformative. What do we know about determination of gynomonoecy exactly ? The statement that it depends on both “genetic and environmental factors” could be true for any trait in every living species..
The following sentences attempt to explain the mechanisms possibly involved in the maintenance of gynomonoecy. But the authors would need to explain better what has been hypothetized / found. For instance, I don’t see how gm can reduce damage to pistil.
Lines 49-53 is a list of keywords which did not help me to understand what questions had been addressed on gynomonoecy. (and we discover afterwards that the sexual system investigated here is actually not gynomonecy but another one, with the co-occurrence of hermaphroditic and gynomonecious individuals).
About the predictions made by the authors at the end:
I agree that the rate of self pollen may be higher on hermaphroditic flowers compared to pistillate. So, if there is inbreeding depression, we should expect progeny performance in open treatments to be higher in female compared to hermaphroditic flowers. It may be what the authors say at the end of the intro, but it seems quite different to what they say at the beginning of the discussion (where they focus on outcross progeny). I don’t see how “life history traits” of female flowers (what is this exactly ?) can differ.. One possibility is that female flowers have no investment in pollen production, so if resource reallocation occurs at this scale, they could produce better seeds. But crossing this with inbreeding depression is not understandable to me. In the discussion, the author say that the ID is higher” in female flowers. What does this mean ? If more outcross pollen arrives on female flowers, comparing outcross and open in these flowers should lead to a smaller difference. Comparing open and selfed should lead to larger differences but I cannot see why the differences between outcross and self should be different across flower types..
Material and Methods
Among the inconsistencies that made the reading difficult:
Line 101 we read about 300 H plant and 300 GM plants. Whereas we find 600 GM plants in fig. 1.
Line 102: we read about three groups of H plants and three groups of GM plants and I could find what happened to these 3 groups (and I can not find them in Fig. 1)
The code used to describe the combination of plant type/flower type is different between the figure 1 and the legend of the same figure.
The term “pollination type” is sometimes used to describe the 3 pollination treatments, sometimes to describe the combination of pollination type x plant type x flower type (see line 163 : it is referred to 9 levels of pollination type). PLEASE be consistent.
E:S ratio : I am not sure to see what is the difference between initial and final. Both appear in Table 1 but they are not listed in the legend!
Results
Did you check / correct for the correlation between seed number and seed size at the fruit / plant level ?
Seedling biomass and survival : there is an effect of plant type and flower type but I could not find which type was better in any case.
Figure 6: I think there is a mistake here. I cannot find the hermaphroditic flowers of hermaphroditic plants on the graph.
Reviewer 2 Report
Evolution of gynomonoecy has been attributed to ehance outcrossing, avoid interference of male-female function, act as a defense against herbivores, increase attractiveness to pollinators and permit flexible resource allocation. In Eremurus anisopterus, a desert ephemeral, sexual system of this species was identified be hermaphrodite-gynomonoecy. To examine the role of female flowers in Eremurus anisopterus, this study compared the level of inbreeding depression of perfect flowers on hermaphroditic and gynomonoecious plants, and female flowers on gynomonoecious plants. The results showed that, in comparison with outcrossing, selfing resulted in a decrease in all traits, particularly for for female flowers under water stress, indicating the advantage of favoring outcrossing for female flowers. Generally, the manuscript was well organized and presented. I have several suggestions that could be found in the marked PDF file.

Reviewer 3 Report
This MS investigated the reproductive consequences of different pollination modes in a desert herb with hermaphrodite-gynomonoecious sexual system. In particular, the authors examined a suit of early life history traits of the offspring produced by pistillate flowers, hermaphroditic flowers in both plant types under stressful environmental settings. I found this study has clear research questions, well-designed methods, sound analysis and discussions. Hence, I recommend acceptance of the ms. Below are some minor suggestions.
Line1-4, I don’t think the title was attractive enough. How about ‘Female advantage in reproduction and early life history stages of a cold desert perennial ephemeral with hermaphrodite-gynomonoecious sexual system’?
Line 30, delete ‘because’
Line 33-35, In recent decades, evolutionary biologists have great interests in the evolution and maintenance of diverse sexual systems because they might present important mechanism in promoting outcrossing and vital sources of genetic variation.
Line 64, please add the family name
Line 84-86, Flowering begins in late April to early May. (period) Flowers are nectarless and mainly pollinated by halictid bees and honeybees, which usually fly upwards when visiting receme inflorescences. Thus, pistillate flowers…
Line 96 change to ‘with ~64% occurring in…’
Line 111-113, why is the paragraph here?
Figure 3, what does the figure imply? Lower temp induces more embryo growth? Is it a coincidence or general pattern?
Line 305-317, I don’t think this section gives a lot more information. One can tell the relative importance by reading the F values in tables, right?
Line 430-432, it’s interesting that pistillate flowers had significantly higher ID than hermaphroditic ones. Any ideas why?? Especially, the selfed and outcrossed progeny of GmP and GmH probably share the same gene composition. Why do they exhibit different phenotypes? Could all the female advantages discovered in this study attributed to larger sizes of female flowers in larger plants?
